# PR-CAD: PROGRESSIVE REFINEMENT FOR UNIFIED CONTROLLABLE AND FAITHFUL TEXT-TO-CAD GENERATION WITH LARGE LANGUAGE MODELS

## ABSTRACT

The construction of CAD models has traditionally relied on labor-intensive manual operations and specialized expertise. Recent advances in large language models (LLMs) have inspired research into text-to-CAD generation. However, existing approaches typically treat generation and editing as disjoint tasks, limiting their practicality. We propose PR-CAD, a progressive refinement framework that unifies generation and editing for controllable and faithful text-to-CAD modeling. To support this, we curate a high-fidelity interaction dataset spanning the full CAD lifecycle, encompassing multiple CAD representations as well as both qualitative and quantitative descriptions. The dataset systematically defines the types of edit operations and generates highly human-like interaction data. Building on a CAD representation tailored for LLMs, we propose a reinforcement learning–enhanced reasoning framework that integrates intent understanding, parameter estimation, and precise edit localization into a single agent. This enables an "all-in-one" solution for both design creation and refinement. Extensive experiments demonstrate strong mutual reinforcement between generation and editing tasks, and across qualitative and quantitative modalities. On public benchmarks, PR-CAD achieves state-of-the-art controllability and faithfulness in both generation and refinement scenarios, while also proving user-friendly and significantly improving CAD modeling efficiency. The code and dataset are be available at <will be filled in upon acceptance>.

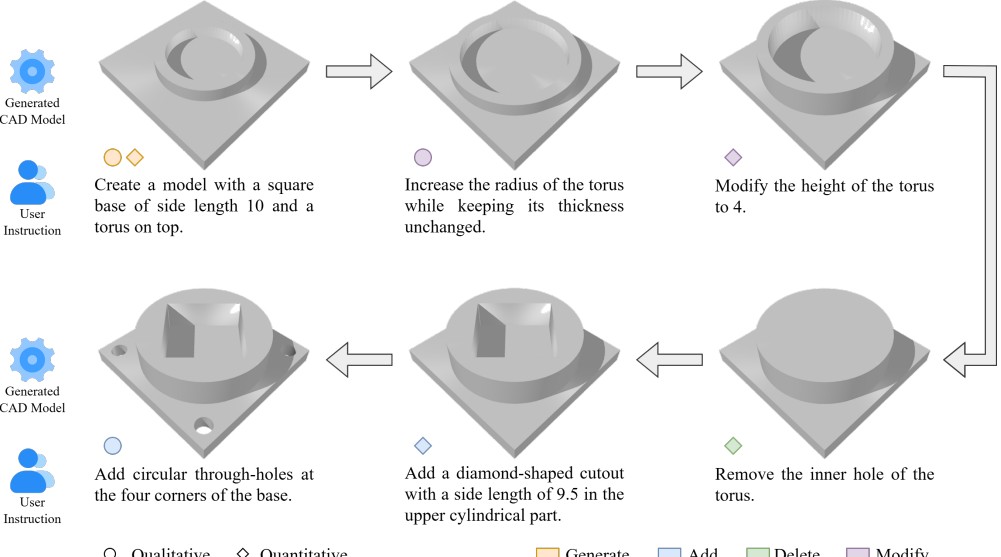

Figure 1: PR-CAD enables user-friendly and controllable CAD generation through progressive refinement.Below each subfigure is the user's intended input described via text instructions, with the corresponding CAD model output above. PR-CAD allows users to generate and progressively refine (add, modify, and delete) their designs from scratch. Users can flexibly choose either qualitative or quantitative descriptions of their intent until the model matches their envisioned design.

# 1 INTRODUCTION

Computer-Aided Design (CAD) is a cornerstone of modern manufacturing, engineering, and industrial design, enabling the creation of precise and complex 3D models. However, traditional CAD modeling remains a labor-intensive process that demands significant expertise, with designers needing to master intricate software interfaces and possess a deep understanding of geometric modeling principles. This steep learning curve limits accessibility and efficiency, highlighting a long-standing research challenge: enabling the intuitive creation and manipulation of CAD models through natural language.

The advent of large language models (LLMs) has spurred progress in text-to-CAD generation, with recent work demonstrating the feasibility of converting textual descriptions into CAD operations. Despite these advancements, a critical limitation remains: current approaches largely treat generation and editing as separate, disjoint tasks. In practice, design is an inherently iterative process. Designers rarely produce a perfect model on their first attempt; rather, they refine their designs through a series of edits (modifying, adding, or removing) features in response to evolving requirements or identified improvements. Existing systems lack a unified framework for this iterative refinement, forcing users to either settle for suboptimal initial models or revert to manual editing outside of the generative process. This disconnection severely limits the practicality and user-friendliness of text-to-CAD systems.

Moreover, many current methods rely on highly detailed, technical text prompts, which are often unnatural for non-expert users to formulate. While some efforts have begun to explore editing, they are often constrained by training data that predominantly features simplistic, randomly generated edits, failing to capture the nuanced, intent-driven nature of real-world design refinements. There is a pressing need for a unified solution that seamlessly integrates generation with controllable, faithful editing based on both qualitative (e.g., "make the base thicker") and quantitative (e.g., "reduce the radius by 6mm") instructions.

To address these challenges, we propose PR-CAD, a progressive refinement framework that unifies controllable and faithful text-to-CAD generation. Our approach is built on three key innovations: First, we curate a high-quality, human-like interaction dataset that spans the entire CAD lifecycle, systematically defining and generating diverse edit operations alongside both qualitative and quantitative descriptions. Second, we introduce a reinforcement learning-enhanced reasoning framework that integrates intent understanding, parameter estimation, and precise edit localization into a single agent, enabling an "all-in-one" solution for both generation and iterative refinement. Third, we employ a structured chain-of-thought (SCoT) methodology to guide the LLM's reasoning, breaking down complex tasks into manageable steps for robust and interpretable generation.

Extensive experiments demonstrate that PR-CAD achieves state-of-the-art performance on public benchmarks, significantly outperforming existing methods in both generation and editing tasks across multiple metrics, including geometric accuracy (Chamfer Distance) and faithfulness to user intent (VLM-Eval). Crucially, human evaluations confirm that our progressive refinement paradigm dramatically improves usability and success rates for both expert and novice users, making professional-grade CAD modeling more accessible.

In summary, our contributions are:

- We propose PR-CAD, a novel progressive refinement framework that unifies text-driven CAD generation and editing within a single, controllable agent, providing a seamless workflow for both creation and modification tasks.

- We introduce a high-fidelity interaction dataset for the full CAD lifecycle, supporting diverse edit types, descriptive modalities, and ensuring comprehensive coverage of CAD interactions from start to finish.

- We develop a reinforcement learning-enhanced reasoning method combining supervised fine-tuning (SFT), structured chain-of-thought (SCoT), and reinforcement learning (RL) to optimize for geometric fidelity, executability, and alignment with user intent, validated through empirical experiments using ChatCAD.

## 2 RELATED WORK

**CAD Model Generation**. Traditional CAD modeling is a highly manual process, requiring deep domain expertise and precision in design. In recent years, research has emerged that aims to automate or simplify this process, particularly by integrating natural language input for CAD generation. Parametric CAD Monedero, 2000 generation has been an important focus, where CAD models are represented as a sequence of operations (e.g., sketching and extrusion). This approach not only describes the geometric structure of the model but also captures the historical process and evolution of the design. With the introduction of large-scale datasets like DeepCAD (Wu et al., 2021), significant progress has been made in text-to-CAD model generation based on Transformer architectures (Vaswani et al., 2017). Khan et al. (2024) proposed Text2CAD, a Transformer-based framework that effectively generates CAD models from textual descriptions. This was followed by additional works leveraging the DeepCAD dataset, such as Text-to-CADQuery (Xie & Ju, 2025) and GeoCAD (Zhang et al., 2025). GeoCAD introduces local geometric controls into the text-to-CAD generation process, enabling more flexible models with fewer input details. Furthermore, Seek-CAD (Li et al., 2025c) introduces a self-improvement mechanism, where the model iteratively refines its designs based on feedback, improving the accuracy of CAD models and enhancing their applicability in real-world scenarios. Lu et al. (2024), Wang et al. (2025), Liao et al. (2025), Li et al. (2025b) and Li et al. (2024) are also paying attention to this research area. However, these methods heavily rely on detailed and highly technical text inputs, making it challenging for users to write instructions that ensure accurate CAD model generation. For example, in the Text2CAD dataset, an average of 100 words is required to describe a single s-E operation, which significantly impacts the practical usability of these models.

**CAD Model Editing**. To address issues such as localized errors or changing requirements in CAD models, the task of CAD model editing has been proposed. Methods like FLEXCAD (Zhang et al., 2024b) explore the random editing of CAD models, allowing users to modify various model parameters without needing to regenerate the entire design. These techniques increase design flexibility but often fall short in high-precision CAD modeling tasks due to the inherent randomness of the editing process. On the other hand, directed editing approaches, such as CAD-Editor (Yuan et al., 2025), allow users to make controlled, targeted modifications to specific parts of a model. These methods offer a more structured editing approach by optimizing certain design features based on predefined parameters. However, these approaches are still constrained by the data construction process, where most of the editing operations are randomly generated. Consequently, the editing data is dominated by add and delete operations, with only a small proportion of quantitative edits, which limits their practical applicability.

**CAD Model Sequential Representation**. The sequential representation of CAD models has been another significant area of research. DeepCAD introduced a domain-specific language (DSL) that describes SE operations using function definitions to represent CAD models. Text-to-CAD proposed a generalized programming language (GPL) based on the cadquery library in Python to generate CAD models. FLEXCAD used structured text (ST) to represent different hierarchical elements of CAD models. These representation methods have been optimized for specific tasks in their respective studies. However, due to the differences in language and format, conversion between these representations is challenging, and they are rarely interchangeable within the same task or context.

**Inference with Large Language Models**. Inference with Large Language Models. Large Language Models (LLMs) excel not only in natural language tasks but also in complex, structured reasoning such as mathematical problem-solving and code generation. Key methods for adapting LLMs to these tasks include Supervised Fine-Tuning (SFT) (Devlin et al., 2019; Hu et al., 2022), Reinforcement Learning (RL) (Sutton et al., 1998; Mnih et al., 2015; Shao et al., 2024), and Chain-of-Thought (SCoT) prompting Li et al., 2025a. SFT trains LLMs on task-specific data to learn domain patterns. RL further refines outputs based on reward signals to improve quality and alignment. SCoT enhances reasoning by prompting the model to generate explicit step-by-step rationales before answering.In CAD generation, these techniques improve model accuracy and controllability. SFT helps learn basic text-to-CAD mappings, RL optimizes design validity and efficiency, and SCoT supports logical planning of construction steps. The combination of SFT, RL, and SCoT offers a promising approach to overcome the reliance on overly technical inputs in existing methods, enabling more intuitive and precise CAD model generation and editing, laying the groundwork for our proposed approach.

# 3 METHODOLOGY

In this section, we introduce PR-CAD, a unified framework for text-driven CAD generation and editing under a progressive refinement paradigm. We detail the core components of our methodology, including the high-quality, human-like interaction data annotation process (Sec. 3.1), as well as the post-training strategy for the progressive refinement model in CAD generation (Sec. 3.2).

## 3.1 HIGH-QUALITY HUMAN-LIKE INTERACTION DATA ANNOTATION

To support both generation and editing tasks, we create a comprehensive interaction dataset that covers the full CAD lifecycle. This includes generating textual descriptions of CAD designs (both qualitative and quantitative) and annotating them with specific edit operations (deletions, additions, modifications). These annotated datasets are then used to train our model to understand, generate, and refine CAD models through a unified interaction framework.

**For Generation Task**, we utilize the DeepCAD dataset to create a comprehensive collection of textual instructions that describe CAD models. These instructions are categorized into two types: qualitative and quantitative. Qualitative descriptions focus on high-level design attributes, while quantitative descriptions involve specific numerical details. In the process of qualitative description generation, we first render the CAD models from multiple perspectives, producing a set of nine views. These views, along with their corresponding JSON descriptions, are used as input for large vision models (Hurst et al., 2024; Bai et al., 2025). This ensures consistency with the original model while emphasizing the broader design intent. For quantitative descriptions, similar to the method of Khan et al. (2024), we directly input the JSON descriptions into a large language model (Team et al., 2023), which generates the specific operations and precise parameters for each step. Simultaneously, to identify a CAD serialization format that is compatible with large models, we generate multiple representations of each CAD model using code translation tools and large language models. Specifically, we employ LogoUp 3D, structured text used by Zhang et al. (2024b). and Python, to represent Domain-Specific Language (DSL), Structured Text (ST) and General-Purpose Language (GPL), as shown in Fig. 2.

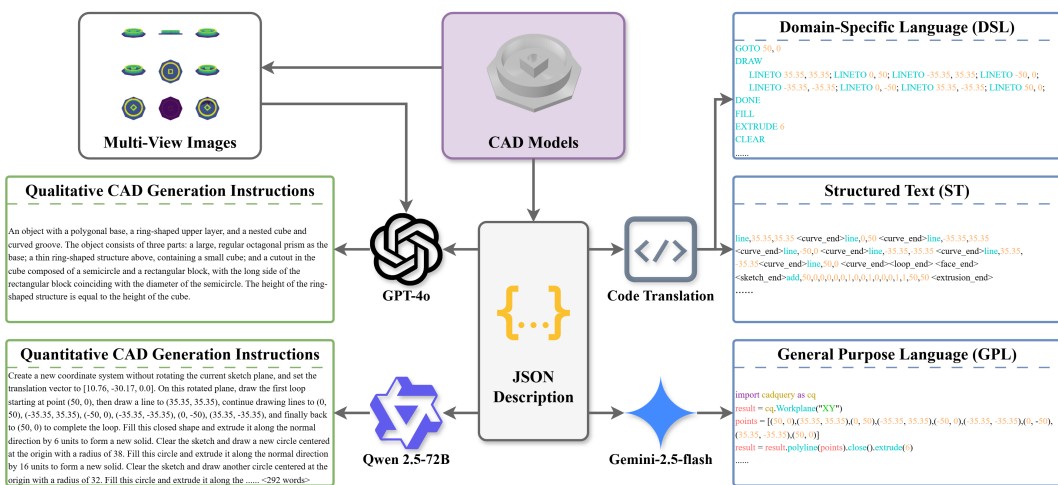

Figure 2: High-quality data annotation pipeline for generation task. Based on the DeepCAD dataset, it generates both generation instructions and CAD sequence representations. Multimodal large models are utilized to produce quantitative and qualitative CAD generation instructions from nine views and JSON descriptions. Code translation and large language models are applied to convert JSON descriptions into various CAD sequence representations.

**For Editing Task**, we developed a multi-stage method for generating interaction data. Specifically, we simulate human deletion actions by systematically removing specific S-E operations or loops from existing CAD models, with the reverse process representing addition actions. Using these CAD model pairs, we train an interaction model capable of both addition and deletion, as outlined in Section 3.2. Furthermore, we generate qualitative addition instructions based on the model obtained after deletion and the removed parts, following the approach shown in Figure 2. Next, we employ

these editing instructions to guide the model in generating new CAD models. Since the model is trained to align with human generation patterns and the newly created parts are derived from those previously created by humans, the new models and the originals form pairs that simulate human editing actions in CAD. The process of creating this interaction data is visually represented in Figure 3, which illustrates the entire workflow: generating CAD model pairs, annotating them with both qualitative and quantitative instructions, and converting them into final editable CAD sequence representations. This multi-stage annotation process ensures that our model can learn a broad spectrum of human editing operations.

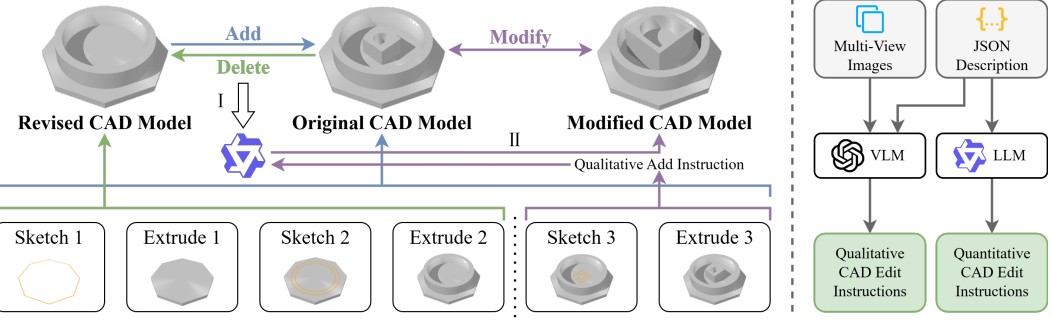

Figure 3: Human-like instruction annotation pipeline for CAD model editing task. In the first stage, CAD model pairs are generated by removing S-E operations or loops, forming deletion and addition pairs as training data (as shown in I). Simultaneously, qualitative add instructions are generated based on the deleted portions. Next, using the trained add/delete model, new CAD models are generated based on the above instructions (as shown in II). The resulting new model and the original model form the edited CAD model pair. Finally, the JSON descriptions are converted into editing instructions and sequence representations in the same manner as for generation tasks (Figure 2).

## 3.2 PROGRESSIVE REFINEMENT CAD GENERATION MODEL POST-TRAINING

In the following, we will detail how we fully leverage the potential of large language models through post-training techniques to achieve progressive refinement for CAD generation. First, we introduce a structured chain of thought (SCoT) method using structured text (see Fig. 4(a)). By combining supervised fine-tuning (SFT) with SCoT data, we enable the LLM to generate CAD models using domain-specific language and enhance its understanding of reasoning patterns(see Fig. 4(b)). Finally, we incorporate reinforcement learning based on Generalized Reward Optimization (GRPO) to further improve the model's generalization capabilities(see Fig. 4(c)).

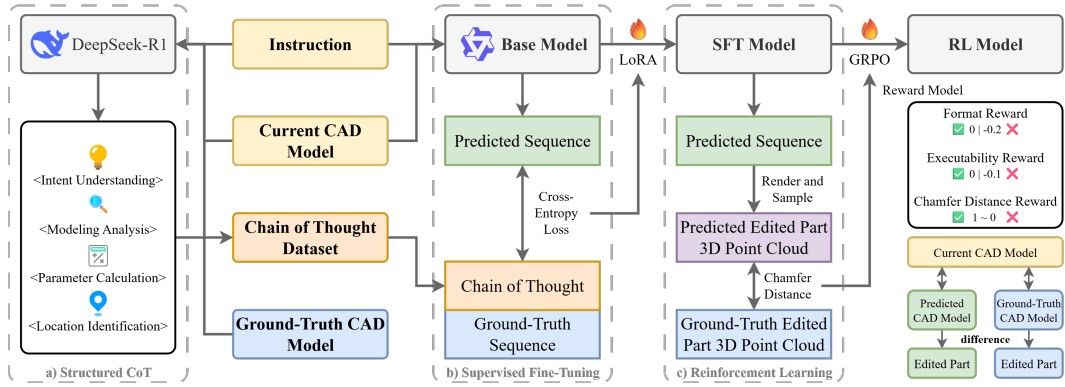

Figure 4: Overview of the PR-CAD post-training process. The post-training process consists of two stages: supervised fine-tuning and reinforcement learning. Using the highly human-like interaction data curated earlier, we employ Qwen2.5-7B-Instruction as the base model. First, DeepSeek (Liu et al., 2024) is used to construct a structured chain of thought by analyzing the input and output data. Then, the model is fine-tuned through supervised learning with cross-entropy loss to learn the task patterns. Afterward, reinforcement learning is applied to the fine-tuned model using rewards based on chamfer distance, among others. For generation tasks, the Current CAD Model is initially empty; for editing tasks, chamfer distance is calculated only for the edited portion of the model.

We found that the performance of LLMs varies across different tasks, depending on the CAD serialization method employed. Specifically, domain-specific languages (DSLs) produce the best results for generation tasks, while structured text proves most effective for editing tasks. The complete results can be found in Appendix A.1. Therefore, to achieve better generation performance, we construct a structured chain of thought using structured text and select a domain-specific language (LogoUp 3D) as the model output in the subsequent experiments (Wong et al., 2025).

**Structured Chain of Thought (SCoT)** guides the model's reasoning process through structured text formats, breaking down the design process into four key steps: intent understanding, modeling analysis, parameter computation, and position identification. In the intent understanding step, the current CAD model is described in textual form and aligned with the user's instructions to match the intended design. Modeling analysis uses special markers (Wang et al., 2023), such as <sketch>, </sketch>, to represent elements in the CAD model and their relationships. Parameter computation performs necessary calculations for transformations like coordinate system rotation, sketch plane displacement, and arc parameters, ensuring the generated model matches the desired geometric properties. Finally, position identification identifies the portions of the CAD model that need to be edited and outputs the corresponding sequence of actions. To support this process, we use a dataset of 1,000 triplets consisting of current CAD models, textual instructions, and target CAD models, which are fed into the DeepSeek-R1-671B model to generate the structured reasoning chain, forming the reasoning chain dataset.

**Supervised Fine-Tuning (SFT)** plays a crucial role in refining the LLM's ability to generate CAD models by leveraging a cross-entropy loss function. Through SFT, the model learns the patterns and relationships inherent in the task by mapping input instructions to the corresponding domain-specific language for CAD modeling. This step involves fine-tuning a pre-trained LLM using structured reasoning data, allowing it to develop a deeper understanding of CAD generation and improve its predictive capabilities.

**Reinforcement Learning (RL)** process depends crucially on an efficient reward function (Devidze et al., 2024; Xie et al., 2023; Devidze, 2025). Our reward function is designed to guide the reinforcement learning agent toward generating high-quality geometric shapes by optimizing for several key properties: format correctness, executability, geometric accuracy, and output length. The total reward $R$ is a summation of four distinct reward components:

$$R = R_{\text{chamfer}} + R_{\text{format}} + R_{\text{exec}} + R_{\text{length}} \tag{1}$$

- Chamfer Distance Reward ($R_{\text{chamfer}}$): This is the primary reward component, a dense reward that measures the geometric similarity between the generated shape and the ground truth. We use the Chamfer Distance ($D_{\text{CD}}$) as our metric. To map this distance to a reward value between 0 and 1, we use an exponential decay function. This ensures that even small improvements in geometry are rewarded, with the maximum reward of 1 given for a perfect match ($D_{\text{CD}} = 0$) (Guo et al., 2025)..

$$R_{\text{chamfer}} = e^{-\alpha D_{\text{CD}}} \tag{2}$$

where $\alpha$ is a hyperparameter controlling the decay rate.

- Format Reward ($R_{\text{format}}$): This is a sparse reward that provides a significant penalty if the agent's output does not conform to the predefined format specifications. A valid output receives no penalty, while an invalid one is penalized by $-0.2$.

$$R_{\text{format}} = \begin{cases} 0 & \text{if format is correct} \\ -0.2 & \text{if format is incorrect} \end{cases} \tag{3}$$

- Executability Reward ($R_{\text{exec}}$): This component penalizes outputs that are not executable or lead to runtime errors. A penalty of $-0.1$ is applied if the generated sequence of actions or code fails to execute successfully.

$$R_{\text{exec}} = \begin{cases} 0 & \text{if executable} \\ -0.1 & \text{if not executable} \end{cases} \tag{4}$$

- Length Reward ($R_{\text{length}}$): To encourage the agent to find concise and efficient solutions, we introduce a penalty proportional to the length of the generated output sequence, $L$. This prevents the agent from generating unnecessarily long or complex outputs to achieve the same result.

$$R_{\text{length}} = -\beta \cdot L \tag{5}$$

where $\beta > 0$ is a hyperparameter that scales the penalty for each added step or token (Ling et al., 2025).

# 4 EXPERIMENTS

## 4.1 EXPERIMENTAL SETUP

**Datasets.** Earlier CAD generation research was constrained by the Transformer architecture, which could only represent specific model parameters within a fixed numerical range, leading to the scaling of real parameters to a predefined interval. In the era of large models, this limitation no longer applies, and we found that excessive scaling might introduce unnecessary errors. Therefore, as described in Section 3.1, we have reconstructed a text-to-CAD model dataset based on the DeepCAD dataset that does not rely on scaling. We also randomly split the dataset into training, validation, and test sets with a 90%-5%-5% ratio.

**Implementation Details.** We use the open-source Qwen 2.5-7B-Instruct as the base model. During Supervised Fine-Tuning (SFT), we employ the LLaMA-Factory framework (Zheng et al., 2024)and apply LoRA with a rank of 8 is applied across all layers, and a sequence length cutoff of 4096 tokens is set. Training runs for 3 epochs with a batch size of 4 per device and 8 gradient accumulation steps. The cosine learning rate scheduler is configured with a learning rate of 1.0e-4 and a warmup ratio of 0.1. Training utilizes BF16 precision with a LoRA dropout rate of 0.1. For reinforcement learning, we use the GRPO method from the veRL framework (Sheng et al., 2024), setting $\alpha = 5.0$ and $\beta = 0.01$. The total reward is computed at the end of each generation episode. All experiments are conducted on 8 NVIDIA H20 and 8 NVIDIA L40s. To ensure consistency with prior work, evaluation metrics were computed using the Text2CAD scripts.

**Metrics.** To evaluate the performance of PR-CAD, we employ the following metrics: (1) Mean Chamfer Distance (Mean CD): Measures the geometric similarity between the generated and ground truth models; lower values indicate higher accuracy. (2) Invalidity Ratio (IR): Proportion of invalid models; lower values reflect better reliability. (3) Qwen 2-VL-72B-Instruct (Team, 2024)(VLM-Eval): Assesses how well the model preserves user intent and design expectations using a multimodal large model Lin & Chen (2023); Gu et al. (2024); Zhang et al. (2024a). (4) Human-Eval: Expert evaluation of the quality and relevance of generated models, scored as 1 if the model meets the instruction, otherwise 0.

## 4.2 MAIN RESULTS

We evaluate PR-CAD through comprehensive experiments comparing it to existing methods in both generation and editing tasks. Our results demonstrate that PR-CAD significantly outperforms other state-of-the-art models in terms of controllability, faithfulness, and overall performance across multiple evaluation metrics.

**Comparison with Existing Methods.** PR-CAD outperforms other models in both quantitative and qualitative evaluations, excelling in generation and editing tasks while effectively preserving user intent and design expectations. We evaluated 2,000 CAD models, and Human-Eval scored 52.94.

Table 1: Comparison of PR-CAD with existing methods for both generation and editing tasks. The evaluation metrics include quantitative measures (IR, Mean CD) and qualitative evaluations (IR, VLM-Eval). PR-CAD significantly outperforms other methods across both tasks, demonstrating superior controllability and model faithfulness. CD values are scaled by $10^3$. "✗" indicates that the method does not support the corresponding task. ↑: the higher, the better; ↓: the lower, the better. Best performance is highlighted in **bold**.

| | Generation Task | | | | Editing Task | | | |
|---|---|---|---|---|---|---|---|---|
| **Method** | Quantitative | | Qualitative | | Quantitative | | Qualitative | |
| | IR ↓ | Mean CD ↓ | IR ↓ | VLM-Eval ↑ | IR ↓ | Mean CD ↓ | IR ↓ | VLM-Eval ↑ |
| GPT-4o (zero-shot) | 74.22 | 133.52 | 66.48 | 55.85 | 25.81 | 23.30 | 27.76 | 61.01 |
| GPT-4o (few-shot) | 55.95 | 77.49 | 49.24 | 58.91 | 15.47 | 15.52 | 13.47 | 66.18 |
| Text2CAD | 0.97 | 27.68 | 3.41 | 66.35 | ✗ | ✗ | ✗ | ✗ |
| Text-to-CadQuery | 6.62 | 11.32 | ✗ | ✗ | ✗ | ✗ | ✗ | ✗ |
| FLEXCAD | ✗ | ✗ | ✗ | ✗ | ✗ | ✗ | 18.26 | 64.38 |
| CAD-Editor | ✗ | ✗ | ✗ | ✗ | 7.18 | 8.85 | 5.77 | 69.52 |
| PR-CAD (Ours) | **0.62** | **5.87** | **1.52** | **69.26** | **0.91** | **6.30** | **1.71** | **77.83** |

In the generation task, PR-CAD outperforms GPT-4o, Text2CAD, and Text-to-CadQuery, achieving the lowest Invalidity Ratio (IR) of 0.62 and Mean Chamfer Distance (CD) of 5.87, indicating high geometric accuracy and reliability. In the editing task, PR-CAD continues to outperform existing methods, including FLEXCAD and CAD-Editor, with an IR of 0.91 and Mean CD of 6.30, while also achieving a remarkable VLM-Eval score of 77.83, demonstrating its ability to preserve user intent and design expectations. A detailed comparison in Table 1 reveals that PR-CAD consistently delivers the best performance across both generation and editing tasks, outperforming zero-shot and few-shot variants of GPT-4o, as well as other specialized CAD models. The improvements in IR and Mean CD highlight PR-CAD's ability to generate and refine CAD models with greater precision and fewer errors. The visual comparison of results is shown in Figure 5.

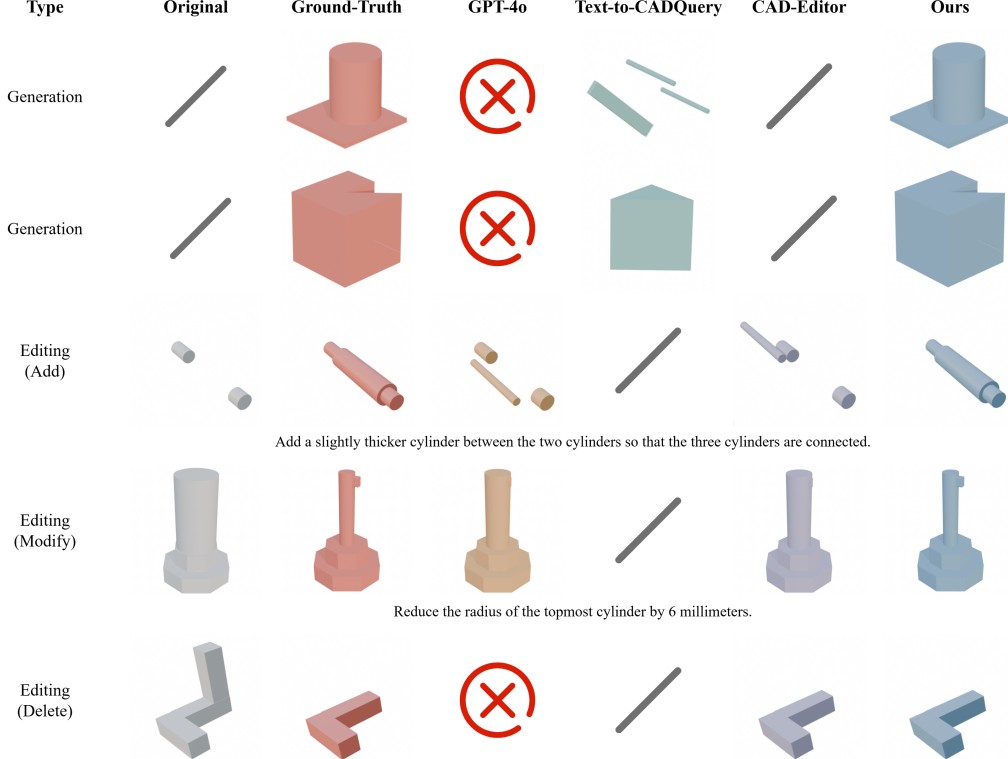

Figure 5: Visualization comparison among different methods or models, including the closed-source model GPT-4o relying on context learning capabilities, Text-to-CADQuery for CAD model generation, CAD-Editor designed for CAD editing, and our proposed method.

Table 2: Human Interaction Results. Comparison of experts and novices using two interaction methods: end-to-end generation and Progressive Refinement. Metrics evaluated include success rate, average number of turns, average time per turn, and System Usability Scale (SUS) scores.

| Interaction Method | Experts | | | | Novices | | | |
|---|---|---|---|---|---|---|---|---|
| | Success Rate | Avg. Turns | Avg. Time | SUS | Success Rate | Avg. Turns | Avg. Time | SUS |
| Single-turn Generation | 56.25% | 1 | 6'48" | 38.125 | 31.25% | 1 | 9'27" | 25.125 |
| Multi-turn Interaction | **100%** | **3.375** | **3'24"** | **93.125** | **81.25%** | **4.53** | **5'16"** | **80.625** |

**User Interaction Performance.** In addition to quantitative performance, we conducted a human evaluation of the CAD models generated by PR-CAD. Table 2 shows that both experts and novices benefit from PR-CAD's multi-turn interaction. Experts achieved a 100% success rate with an average of 3.375 turns and 3'24" per turn. Novices also performed well, with an 81.25% success rate

and improved efficiency compared to single-turn generation. Remarkably, we found that novices outperformed experts using traditional end-to-end methods, with PR-CAD's assistance. The System Usability Scale (SUS) scores (Brooke et al., 1996; Sauro & Lewis, 2011; Lewis, 2018) further underscore PR-CAD's user-friendliness, with experts rating it at 93.125 and novices at 80.625, demonstrating the effectiveness of the progressive refinement approach in enhancing user interaction.

### 4.3 CHATCAD: CAD MODELING THROUGH MULTI-TURN DIALOGUES

To further evaluate the real-world applicability of PR-CAD, we introduce ChatCAD, a system designed for CAD modeling through multi-turn dialogues. As demonstrated in Figure 6, ChatCAD allows users to iteratively refine and update CAD models based on conversational instructions. Our results show that PR-CAD enables seamless transitions between steps, providing high accuracy and flexibility for users to modify their designs through simple dialogue exchanges.

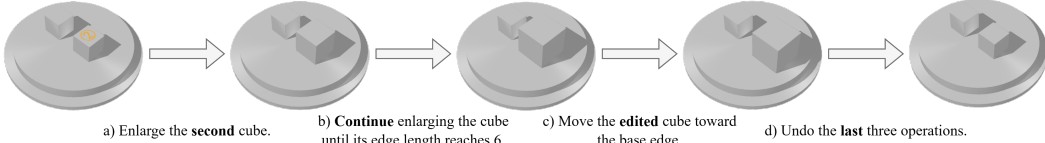

a) Enlarge the **second** cube.   b) **Continue** enlarging the cube until its edge length reaches 6.   c) Move the **edited** cube toward the base edge.   d) Undo the **last** three operations.

Figure 6: Examples of CAD modeling through multi-turn dialogues. (a) In this step, the specified cube is enlarged based on the earlier description. (b) The next step accurately identifies the cube from the previous operation and completes the enlargement. (c) In this step, the edited cube is identified and moved towards the base edge. (d) Finally, this step precisely undoes all the operations performed in the previous three steps.

### 4.4 ABLATION STUDIES

We conduct ablation studies to assess the impact of various components in our model, as shown in Table 3. Our findings indicate that both supervised fine-tuning (SFT) and reinforcement learning (RL) are essential for achieving the best performance. Specifically, removing either SFT or RL leads to substantial drops in performance, particularly in chamfer distance and VLM-Eval scores. Additionally, the structured chain of thought (SCoT) significantly contributes to the model's reasoning capabilities, with its absence resulting in a noticeable decline in model accuracy.

Table 3: Ablation Studies for Post-Tuning LLMs with Various Training Configurations. We assess the impact of different training configurations, including baseline models, reinforcement learning-only fine-tuning (w/o SFT), supervised fine-tuning-only (w/o RL), training without structured reasoning (w/o SCoT), and single-task data training. The notation *t/o/o* (trained only on) indicates that the model was trained exclusively on one type of data. Best performance is highlighted in **bold**.

| Training Strategy | IR | Mean CD | Median CD | VLM-Eval |
|---|---|---|---|---|
| Qwen2.5-7B | ✗ | ✗ | ✗ | ✗ |
| Qwen2.5-7B w/o SFT | ✗ | ✗ | ✗ | ✗ |
| Qwen2.5-7B w/o RL | 12.67 | 84.63 | 6.54 | 56.44 |
| Qwen2.5-7B w/o SCoT | 2.48 | 11.34 | 2.08 | 68.81 |
| Qwen2.5-7B t/o/o Generation | 9.45 | 42.07 | 5.49 | 60.56 |
| Qwen2.5-7B t/o/o Editing | 6.94 | 20.05 | 1.90 | 65.54 |
| Qwen2.5-7B t/o/o Quantitative | 3.71 | 14.75 | 1.72 | 67.19 |
| Qwen2.5-7B t/o/o Qualitative | 7.78 | 34.07 | 2.55 | 66.91 |
| PR-CAD (Ours) | **1.18** | **8.37** | **0.78** | **70.84** |

## 5 CONCLUSION

We present PR-CAD, a unified framework for controllable and faithful text-to-CAD generation and refinement, which integrates both tasks into a single, iterative workflow. Our approach significantly improves performance and usability over existing methods, achieving state-of-the-art results in geometric accuracy, reliability, and user intent preservation. Human evaluations demonstrate its superior usability, especially for novice users, making CAD modeling more accessible. PR-CAD's ability to generate and iteratively refine designs with both qualitative and quantitative instructions offers a powerful tool for democratizing CAD design.

## ETHICS STATEMENT

This research follows ethical guidelines in the use of AI and machine learning technologies. We ensure that all data used is synthetic and anonymized, with no personal or proprietary information involved. Consent was obtained from human evaluators, and ethical standards were maintained throughout the study. Our system was designed to be fair and transparent, with efforts to mitigate biases and ensure that generated designs do not favor any particular group. We aim to democratize CAD modeling, making it accessible to both experts and novices, while acknowledging the potential impacts on traditional CAD roles. We also strive to minimize the environmental impact of our AI models by using energy-efficient hardware and optimizing computational resources.

## REPRODUCIBILITY STATEMENT

To ensure the reproducibility of our work, we will make the full set of resources available upon acceptance of the paper. This includes the source code for the PR-CAD framework, which will be accessible through a public GitHub repository, complete with installation instructions and usage guidelines. We will also release the curated high-quality interaction dataset, encompassing both qualitative and quantitative CAD descriptions along with corresponding edit operations, for academic and research use. Additionally, the pre-trained models used in our experiments will be shared, enabling others to replicate our results directly. All model training details, including hyperparameters, loss curves, and evaluation metrics, are recorded using Weights and Biases (Wandb), and these logs will be made publicly available to ensure full transparency in the training process. We will provide detailed documentation on the experimental setup, including model configurations, hyperparameters, and hardware specifications, to facilitate accurate replication of our results. Through these efforts, we aim to ensure that our research is transparent, reproducible, and accessible to the broader academic community.

## THE USE OF LARGE LANGUAGE MODELS

In order to clarify our work, we used large language models solely for the modification and refinement of a small portion of the written content, focusing on grammar and rhetorical aspects, without involving any substantial content generation. None of the content was generated from scratch by the large language model, nor was any content released without thorough inspection by us after being generated by the model.

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

# Appendix

Due to space constraints in the main paper, additional results and discussions are provided in this appendix, which is organized as follows:

- **Section A: Additional Implementation Details and Analysis.**
  - Sec. A.1: The Impact of Different CAD Serialization Representations
  - Sec. A.2: Errors Caused by Scaling Operations
  - Sec. A.3: Definition of Editing Operation Types
  - Sec. A.4: Injecting Robustness into PR-CAD with Reinforcement Learning
  - Sec. A.5: Limitations and Future Work.

## A  ADDITIONAL IMPLEMENTATION DETAILS AND ANALYSIS.

### A.1  THE IMPACT OF DIFFERENT CAD SERIALIZATION REPRESENTATIONS

In our experiments, we found that the performance of the PR-CAD framework varies based on the serialization method used for CAD models. Specifically, we evaluated different methods such as Domain-Specific Language (DSL), Structured Text (ST), and General-Purpose Language (GPL) for CAD serialization. The results show that DSL-based representations provide the best performance in generation tasks, while ST-based representations are more effective in editing tasks. This distinction is crucial in understanding how CAD serialization impacts both the generation of new models and the iterative editing process. Figure 7 demonstrates the comparative performance across different serialization methods.

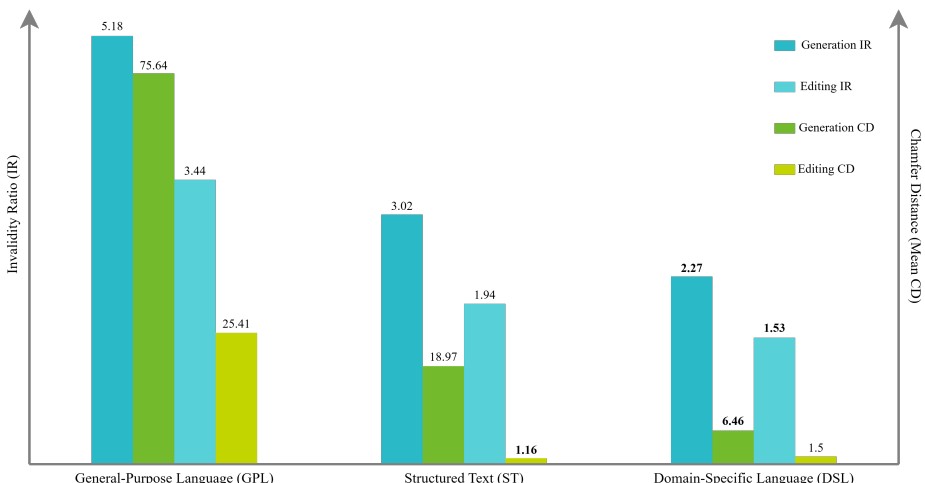

Figure 7: The difference in model performance caused by using different CAD sequence representations, while keeping all other supervised fine-tuning strategies identical. Best performance is highlighted in bold.

While General-Purpose Languages like Python have the advantage of being learned on the largest datasets during training, they are not specifically designed for CAD modeling tasks. As a result, there is a significant gap between their natural language instructions and the CAD representation space, which leads to poor performance in CAD modeling tasks. In contrast, Structured Text, with its embedded structural tags, excels in editing tasks by allowing accurate positioning. However, its excessive parameters and unclear semantics make it less effective for generation tasks. On the other hand, Domain-Specific Languages incorporate part of the CAD sequence structure while being closer to natural language and human cognitive patterns, making them ideal for generation tasks. Based on this analysis, we treat Structured Text as part of a structured reasoning chain and ultimately use Domain-Specific Language to represent CAD models.

## A.2 Issues Caused by Scaling Operations

Scaling operations in CAD model generation can lead to significant issues when applied indiscriminately. In our study, we noted that excessive scaling of model parameters, which was previously a limitation in older transformer-based methods, could introduce undesirable deviations in geometric accuracy. To mitigate these issues, we reconstructed our dataset based on the DeepCAD dataset, avoiding the need for scaling operations and directly using raw CAD model data. This approach significantly reduced issues in model generation, as shown in Figure 9.

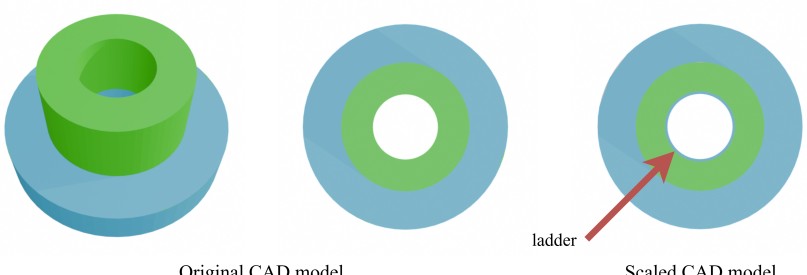

Original CAD model          ladder          Scaled CAD model

Figure 8: Numerical issues due to scaling. The above image shows a typical example where accuracy errors caused by scaling result in internal through-holes, which should have the same radius, displaying ladder.

## A.3 Definition of Editing Operation Types

In this study, we defined several types of editing operations to ensure robust interaction modeling. These operations include:

- Addition: Involves introducing new S-E operations or loops to an existing model.

- Modification: Involves changing the properties or parameters of existing elements.

- Deletion: Involves removing S-E operations or loops from a model.

Each of these operations plays a vital role in the iterative refinement of CAD models. We annotated a diverse set of CAD model interactions, ensuring that each operation type was well-represented in our training data, enabling the model to handle real-world editing scenarios effectively.

## A.4 Injecting Robustness into PR-CAD with Reinforcement Learning

In the model after reinforcement learning, we observed an interesting phenomenon. When the input command contains potential errors or is likely to cause a crash, the model actively adjusts parts beyond the editing intent to prevent the error from occurring. We believe this is related to the Executability Reward used during the reinforcement learning phase.

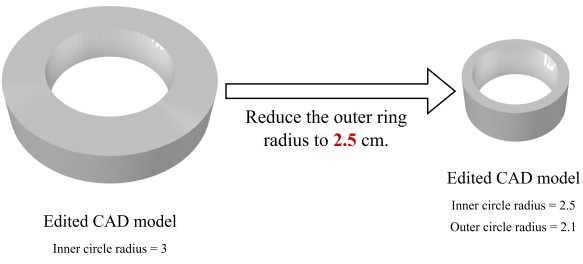

Reduce the outer ring radius to **2.5** cm.

Edited CAD model
Inner circle radius = 2.5
Outer circle radius = 2.1

Edited CAD model
Inner circle radius = 3
Outer circle radius = 5

Figure 9: Robustness Example. When the user requests the outer ring radius to be smaller than the inner ring radius, the model automatically reduces the inner ring radius to ensure the model can generate correctly.

### A.5    LIMITATIONS AND FUTURE WORK.

Despite the promising results, PR-CAD has certain limitations. While our approach successfully integrates CAD generation and editing, strictly adhering to the user's qualitative and quantitative intentions, and performing excellently in interactive modeling within real-world scenarios, it still requires further refinement to handle more complex designs and specialized domains. In future work, we aim to enhance the model's planning capabilities to support more intricate editing operations and further improve its efficiency for large-scale CAD projects. Additionally, we plan to explore more advanced techniques for model scalability, enabling real-time applications in industry-specific scenarios.

