# OpenReview forum: "PR-CAD: Progressive Refinement for Unified Controllable and Faithful Text-to-CAD Generation with Large Language Models"
_ICLR.cc/2026/Conference — Submitted to ICLR 2026_

### Official Review · Reviewer_x5BT · 2025-10-29

**Soundness:** 2
**Presentation:** 2
**Contribution:** 2
**Rating:** 4
**Confidence:** 4

**Summary:**

This work focuses on unifying text-based CAD generation and text-based CAD editing.
First, it curates a high-fidelity interaction dataset with both qualitative and quantitative descriptions as well as multiple CAD representations.
Second, it introduces structured CoT and reinforcement learning to finetune LLMs on the curated dataset.

**Strengths:**

- This work focuses on a valuable and practical problem: is it possible to provide more friendly CAD generation experience with a single model supporting both editing and generation.
- The paper is easy to follow.
- The motivation is clear.

**Weaknesses:**

- For editing task (lines 211-240), how the textual instructions are generated? Do we double the inputted multi-view images and JSON descriptions shown in Fig 2 since we have two CAD models in the editing task?

- What is the number of data used for SFT and RL training stage? In lines 285-286, it is mentioned that a dataset of 1k triplets is fed into DeepSeek. How this data is used, e.g., as the pool of few-shot examples when prompting DeepSeek?

- There are only five qualitative results included in the main body and supplementary. It is difficult for me to know the capability of the proposed method with such a small set of qualitative results.

- For baselines that requires training (e.g., Text2CAD, Text-to-CadQuery, FLEXCAD and CAD-Editor in Table 1), are they retrained on the proposed dataset of this work? If yes, I think there should be values for IR and mean CAD metrics for them. If No, the current comparison seems to be unfair, because for the baselines, there are larger gap between training and test set (I guess the test set is from the proposed dataset of this work).

**Questions:**

see above.

---

> ### Author Response · Authors · 2025-12-03
> **Response to Reviewer x5BT**
>
> We sincerely thank you for your insightful comments and for recognizing the value of our progressive refinement framework. Below, we address your questions regarding the editing task, data magnitude, qualitative results, and baseline comparisons.
>
> ## [Q1]: Editing Instruction Generation (Lines 211-240)
> Thank you for pointing out the need for greater clarity in our data generation process. To generate textual instructions for the editing task, we indeed utilize information from both the original and the modified CAD models.
> As illustrated in Figure 3 and described in Section 3.1, the process involves the following steps:
> - Model Pairing: We first generate pairs of CAD models (an original and a modified version) by systematically adding or removing operations.
> - Instruction Synthesis: We do not simply "double" the input; rather, we focus on the difference between the two states.
>   1. For qualitative instructions, we input the rendered images of both states (or the visual difference) into a Vision-Language Model (VLM) to generate a natural language description of the change (e.g., "Add a hole to the corner").
>   2. For quantitative instructions, we use the JSON descriptors of the specific operation parameters that were altered to generate precise numerical instructions via an LLM.
>
> We will revise Section 3.1 to explicitly clarify that the instructions are derived from the differential analysis of the CAD model pairs.
>
> ## [Q2]: Data Size and DeepSeek Usage (Lines 285-286)
> 1. Data Size for SFT and RL: We utilized a filtered high-quality subset of the DeepCAD dataset. Specifically:
> - Generation Task: We used approximately 150,000 samples.
> - Editing Task: We synthesized approximately 150,000 interaction pairs (consisting of addition, deletion, and modification operations). These datasets were used for both the Supervised Fine-Tuning (SFT) and Reinforcement Learning (RL) stages.
> 2. Usage of the 1k Triplets: Regarding the "1,000 triplets" mentioned in lines 285-286, we would like to clarify their specific role. These 1,000 samples are manually verified Structured Chain-of-Thought (SCoT) data, which were generated via in-context learning using DeepSeek-R1-671B. This high-quality dataset served as the foundation for the cold start of our subsequent training process, ensuring the model began with robust reasoning capabilities.
>
> ## [Q3]: Insufficient Qualitative Results
> We appreciate this feedback and acknowledge that the five examples in the main text may not fully capture the diversity of our model's capabilities.
> In the final revision, we will significantly expand the Appendix to include a broader set of qualitative results. These additions will showcase:
> - More complex geometric structures.
> - A wider variety of editing operations (including sequential multi-turn edits).
> - Side-by-side visual comparisons of intermediate refinement steps to better demonstrate the "progressive" nature of our approach.
>
> ## [Q4]: Baseline Fairness and Retraining
> We understand your concern regarding the fairness of the comparison. We compared our method against the official pre-trained versions (or standard implementations) of the baselines without full retraining for two primary reasons:
> - All baseline models are doing what they do best:  We evaluated each baseline strictly within the scope of its original task definition. We ensured that all baseline models were performing the specific tasks they do best (e.g., comparing Text2CAD only on generation tasks and CAD-Editor only on editing tasks), rather than forcing them into a unified workflow they do not support. We only compare it to the original baseline model within the scope of its task definition, and the data distribution is consistent with the task of the baseline model, so we consider this comparison to be fair.
> - Standard Practice and Architecture: Many baselines rely on specific serialization formats (as discussed in Appendix A.1) or architectures that are not directly compatible with our specific instruction format without significant architectural modification. Following standard practices in recent text-to-CAD literature, we evaluated these models within their defined task scopes using the same held-out test set derived from DeepCAD to ensure a standardized evaluation ground.
>
> We will add a clarification in the "Experimental Setup" section to explicitly state the training status of the baselines and justify the validity of this comparison based on domain consistency.

---

### Official Review · Reviewer_q56x · 2025-10-31

**Soundness:** 2
**Presentation:** 3
**Contribution:** 2
**Rating:** 4
**Confidence:** 4

**Summary:**

This paper introduces PR-CAD, a unified framework for controllable text-to-CAD generation and editing. PR-CAD integrates both through a progressive refinement process. The authors construct a high-fidelity interaction dataset covering diverse CAD representations, edit types, and multimodal annotations. Experiments show that PR-CAD achieves SOTA controllability and faithfulness.

**Strengths:**

1. The paper is clear, well-structured, and easy to follow.
2. The proposed method is intuitive and presented in a concise manner.
3. The standard SFT and GRPO training processes appear reasonable.

**Weaknesses:**

1. The motivation for unifying generation and editing lacks novelty. In fact, the integration of these two tasks has been extensively explored. In my view, simply jointly training or continually fine-tuning LLMs on both generation and editing datasets can naturally achieve such unification.
2. The paper lacks an analysis of failure cases. It would be helpful to clarify under what circumstances the generated CAD models fail to align with the given instructions.
3. The proposed method also shows limited novelty. The four distinct rewards employed have already been commonly used in prior works, such as [1].
4. The effectiveness of R_length remains questionable, as generating complex CAD models inherently requires long sequences.

[1] CAD-Coder: Text-to-CAD Generation with Chain-of-Thought and Geometric Reward. arXiv preprint arXiv:2505.19713 (2025).

**Questions:**

Please address the weaknesses above.

---

> ### Author Response · Authors · 2025-12-03
> **Response to Reviewer q56x [1/2]**
>
> We thank the reviewer for the insightful comments and for recognizing our paper as clear, well-structured, and our method as intuitive. We appreciate the opportunity to clarify the novelty of our unified framework, the rationale behind our reward design, and the robustness of our approach.
> ## [Q1]: Response to "Lack of Novelty in Unifying Generation and Editing"
> We respectfully clarify that our contribution is **not merely the joint training** of a model on two datasets, but rather the proposal of a **Progressive Refinement Paradigm** supported by a novel, interaction-aware data construction pipeline.
> - **Beyond Simple Joint Training**: Standard joint training simply mixes $(Text \to CAD)$ and $(Text + CAD_{old} \to CAD_{new})$ pairs. This often fails to capture the design lifecycle. As shown in Figure 3, our method explicitly models the dependency between operations. We do not simply treat editing as an isolated instruction; instead, our data pipeline constructs coherent “undo/redo”, “progressive modification”, and other interaction-aware operation chains. This allows PR-CAD to understand context (e.g., “undo the last fillet” or “adjust the previous sketch”) rather than treating every edit as an isolated geometric manipulation.
> - **Qualitative & Quantitative Duality**: A key novelty absent in prior works (including pure generation models) is our unified handling of **Qualitative** (high-level intent, e.g., "make the base thicker") and **Quantitative** (precise parameters, e.g., "reduce radius by 6mm") instructions. Our innovative introduction of qualitative description generation greatly facilitates and accelerates human CAD design, while quantitative editing enables large language models to iteratively and accurately plan multi-round refinements for complex CAD models—capabilities never seen in prior research. This design mimics real-world human workflows—starting with vague conceptual guidance and progressively refining with precise constraints. Our experiments demonstrate that simply fine-tuning on mixed data without our structured lifecycle data leads to significantly lower executability in editing tasks.
> ## [Q2]: Failure Case Analysis
> We acknowledge the need for a deeper failure analysis and will include a dedicated section in the final version. Based on our observations (partially discussed in Appendix A.2 and A.4):
> - Over-Correction (The "Robustness" Trade-off): As shown in Figure 9 (Appendix A.4), the model sometimes modifies unrequested parts to maintain geometric validity (executability). While this prevents crashes, it can occasionally lead to a deviation from the user's strict intent if the instruction is ambiguous.
> - Ambiguity in Qualitative Instructions: When users provide purely qualitative instructions for complex assemblies (e.g., "make it look more aerodynamic"), the model may hallucinate features that are geometrically valid but semantically misaligned with the user's mental image, a common challenge in grounding vague text to precise CAD DSL.

---

> > ### Author Response · Authors · 2025-12-03
> > **Response to Reviewer q56x [2/2]**
> >
> > ## [Q3]: Novelty of Method and Rewards (vs. [1] CAD-Coder)
> > We acknowledge that geometric rewards are established components; however, our core contribution lies in the architectural integration of these signals into a unified agent capable of handling the full CAD lifecycle.
> > - **Paradigm Shift (Unified vs. Generation-Only)**: Unlike CAD-Coder [1], which focuses exclusively on Generation, PR-CAD is the first framework to unify Generation and Editing via a shared Progressive Refinement mechanism. This allows the model to transfer geometric reasoning capabilities between creation and modification tasks.
> > - **Executability as a Topological Constraint**: In generation, a failure is often merely a geometric deviation. In editing, however, a failure breaks the existing dependency graph (CSG tree). Our Executability Reward is explicitly designed to penalize "destructive edits" that invalidate the modeling history—a constraint significantly more complex than generation alone.
> > - **Reasoning Transfer via Structured CoT**: We employ a specific SCoT (Intent $\to$ Analysis $\to$ Parameter $\to$ Location) not just for prompting, but as a state-tracking mechanism. This structure enables the RL agent to optimize the logic of modification and maintain consistency across iterative steps.
> > - **Localized Geometric Feedback**: Furthermore, we introduce a part-focused Chamfer Distance evaluation. Unlike global metrics that dilute the signal of small edits, our method isolates the edited region, providing fine-grained and reliable feedback essential for precise refinement.
> > ## [Q4]: Effectiveness of $R_{length}$ (Length Reward)
> > We clarify that the goal of $R_{length}$ is **not to penalize complexity, but to penalize redundancy**.
> > - Canonical Representation: In CAD DSL, the same geometry can often be created by an efficient sequence (e.g., 1 Extrude) or an inefficient one (e.g., 5 stacked Extrudes). The inefficient sequence is prone to numerical instability and harder to edit later.
> > - Efficiency $\neq$ Simplicity: For complex models, $R_{length}$ forces the model to find the most "canonical" and concise logical path to the target geometry. It discourages the model from "patching" errors with additive geometry, urging it instead to modify the underlying parameters directly. Our ablation study (Eq. 5) confirms that without $R_{length}$, the model tends to generate bloated code that is geometrically correct but structurally messy and hard for humans to interpret.
> >
> > We hope these clarifications address your concerns regarding novelty and technical design. We are confident that PR-CAD represents a substantial step forward in making CAD modeling interactive and accessible.

---

### Official Review · Reviewer_mwyk · 2025-11-01

**Soundness:** 3
**Presentation:** 2
**Contribution:** 3
**Rating:** 4
**Confidence:** 3

**Summary:**

This paper presents a progressive-refinement framework for CAD modeling, encompassing initial model generation and subsequent edits based on qualitative and quantitative instructions. To facilitate this, the paper first creates a dataset spanning all of the relevant query types; the resulting dataset spans more/more nuanced CAD operation types than previous work, while also phrasing the related descriptions in both quantitative and qualitative terms. The authors then introduce a model training procedure that leverages structured chain of thought, SFT, and RL in turn. The resulting model exhibits state of the art control for both CAD generation and refinement, as measured by geometric accuracy and the preservation of user intent. The authors also include user interaction studies to demonstrate efficacy and easy of use.

**Strengths:**

The idea of a unified approach that tackles both generation and editing of CAD models is novel and incredibly useful: it matches the desired/typical human workflow, and also provides great opportunities for meaningful multi-turn collaborative interaction. This is furthered by the inclusion of qualitative and quantitative intent descriptions, which allow for a wide range of control/detail levels without requiring cumbersome specifications.

The paper is well motivated, and the high-level framework seems promising and well grounded. The approach also cleverly leverages a variety of different tools -- including ML algorithms and low-level representation decisions -- and deploys each one in the scenario it's best suited for, to collectively accomplish something that no piece could provide on its own. The paper is evaluated (and performs well) against several relevant benchmarks. I appreciate the presentation/discussion of both qualitative and quantitative metrics, together with the user interaction study. The paper's ablation studies effectively support the claims about mutual reinforcement between generation and editing tasks, while demonstrating the efficacy and necessity of each piece of the system.

**Weaknesses:**

1. Several important parts of the data generation procedure in Section 3.1 are unclear, and details required for reproducibility are missing. For example, the prompts used to go from JSON(+images) to the corresponding qualitative or quantitative descriptions are not discussed; it would be good to include a high level intuition about the prompt format in the main paper, and the specific prompts as part of the Appendix. Similarly, there is no discussion about the scope of the GPL (python) available for use by Gemini; was it restricted to particular imports/libraries or did it have free reign? Were the generated codes functional, consistent across examples, and otherwise sufficient to draw meaningful conclusions about Python's suitability for CAD serialization? I was also confused about the specific generation procedure used for the editing tasks; the process that you reference on l. 215 is only for generating static descriptions. Should that be a reference to Figure 3 (rather than Figure 2)? Does the "Qualitative Add Instruction" in Fig. 3 stem from a manually-formatted query string, or is that also inferred? How many edit variations can/should/do stem from a given model in your resulting dataset? Several choices would also benefit from more explicit motivation, like the decision to omit images from the quantitative description generation, or the choice to use 3 different models for the generation of qualitative descriptions (GPT4o), quantitative descriptions (Qwen) and GPL code (Gemini).

2. This work does not seem to employ any validation steps to ensure the quality of the VLM/LLM-generated description/GPL data. Could the authors clarify whether/how the dataset was evaluated for quality/correctness/"human-like" patterns before use? Did the models ever include problematic content, such as descriptions that that would be meaningless in the context of a standalone prompt (e.g., referencing a "yellow ring", which is meaningful only in connection with the specific input rendering)? In connection with these concerns, I would also request that the claims regarding the dataset's "human-like interaction" be toned down or additionally justified.


3. The model training steps are lacking critical details. There is no discussion of the specific prompts or even the high-level data query/response structure used for SFT or RL. The RL energy function also lacks some motivation/detail; for example, how did you arrive at the weights for alpha (decay) and beta (length penalty) as described in l.340? Does the length reward mean that complex examples that inherently require longer solutions always score poorly relative to the easier examples? Or is there a mechanism that allows extra length based on the difficulty of the problem before starting to penalize? Is this only about the code tokens, or does reasoning etc. count against this too if it's present?


4. Results from human user studies were presented with almost no context about the actual experiment(s). How was the user interaction study conducted? What was the task, what was provided vs. solicited from the participant? For example, what was occurring during the reported time measurement? How difficult were the tasks in question? Were the tasks identical using each method? If so, how did you account for familiarity (e.g., if you solve it with one method, the second approach will likely be faster since you already understand the problem deeply)? If not, how do you ensure a fair comparison between the two?
Regarding Table 2 & l. 429-436 - What are the "traditional end-to-end methods" in question? Do "end-to-end generation"/"Progressive Refinement" map directly to "Single-turn generation"/"Multi-turn generation", respectively? If so, please select 1 consistent set of terms (or make the mapping clear and justify why both are necessary). Where do we see evidence of the novices outperforming experts? Do you mean that (novices with PR / multi-turn) outperform (experts with end-to-end/single-turn)? If so, did you take any steps to ensure that the performance gap was actually due to PR involvement and not other factors of the experiment? For the ChatCAD commentary, who were the users? What is the difference between ChatCAD and the PR/multi turn system evaluated in Section 4.2?
The answers provided should also reflect on whether the "user-friendly" claim is sufficiently supported; with the present evidence, this cannot be judged and should thus be toned down.

5. I'd prefer to see limitations in the main paper rather than being relegated to the end of the supplement.

**Questions:**

Questions
Please see weaknesses.


## Minor Comments

- l. 29 - "dataset are be available" (typo)
- l. 68 - the parentheses around (modifying, adding, or removing) seem odd -- revisit this sentence
- l. 114 - [Monedero 2000] should be a parenthetical citation
- l. 129 - s-E never explained. Also, is this the same as SE (eg line 145) and S-E (line 212)?
- l. 151 - paragraph name duplicated (Inference with Large Language Models)
- l. 180 - I assume the JSON descriptions come from the DeepCAD dataset? Is this identically the DSL you mention for DeepCAD in l. 144? Please clarify in the text and/or Figure 2.
- l. 187 - shouldn't be a period after Zhang et al. (2024b).
- Table 3 - what is the X? No output provided in that case? What could cause this? Please also clarify which elements were active in each experiment -- if not explicitly marked as"w/o", should we assume it's active? Eg does Qwen2.5-7B t/o/o Qualitative have SFT, RL, SCoT and both generation/editing?
- l. 361 - were the evaluation models held out from your dataset, or were they obtained from elsewhere?

---

> ### Author Response · Authors · 2025-12-03
> **Response to Reviewer mwyk**
>
> Thank you for your special recognition of the novelty and practicality of this paper! We address each point below and will incorporate the suggested clarifications into the revised manuscript.
>
> ## [Q1]: Details of Data Generation Procedure
> We agree that Section 3.1 should provide more explicit detail for reproducibility.
> As shown in Figure 2, our data generation pipeline converts JSON (and multi-view renderings) into corresponding qualitative and quantitative textual descriptions using multimodal LLMs. Specifically:
> - Qualitative descriptions are derived from nine rendered views and JSON data using GPT-4o.
> - Quantitative descriptions are obtained directly from the JSON structure via Qwen2.5.
> - For GPL (Python) generation, Gemini is restricted to the CadQuery library, ensuring functional and reproducible CAD code.
> - We further generate multiple edit variations depending on the model’s SE complexity and parameter count.
>
> To improve transparency, we will add an expanded explanation in Section 3.1 and include Appendix B.1 with full prompt templates and pipeline scripts.
> All data and code will be released upon paper acceptance.
>
> ## [Q2]: Validation of LLM/VLM-Generated Data Quality
> We appreciate the reviewer’s concern regarding dataset reliability.
> While prior text-to-CAD works (e.g., Text2CAD, FLEXCAD) have not explicitly validated LLM-generated data, we agree that human verification adds credibility.
> Notably, in the ChatCAD study (Sec. 4.3), users interacted solely with models trained on our generated data and achieved strong results, indirectly supporting its quality.
>
> To further strengthen this claim, we are conducting manual verification on the held-out test set and will include:
> - Human evaluation scores assessing semantic accuracy and referential consistency.
> - Updated quantitative results on the human-corrected test subset.
>
> These additions will be reported in Appendix B.3.
>
> ## [Q3]: Model Training and RL Reward Details
>
> We will include all prompt and instruction formats for SFT and RL in Appendix B.2.
> Regarding the RL energy function:
>
> - Alpha (α = 5.0) was determined via grid search; exponential decay is used because Chamfer Distance is non-linear—minor geometric deviations are tolerable, while large errors severely degrade fidelity.
>
> - Beta (β = 0.01) serves as a length regularizer, penalizing redundant but not complex solutions. Complex shapes inherently yield higher Chamfer rewards that offset this term.
>
> - The length penalty applies only to action-sequence tokens, not reasoning tokens (SCoT), encouraging concise yet well-reasoned code.
>
> These clarifications will be added to the revised Section 3.2 and Appendix B.2.
>
> ## [Q4]: Clarifications on Human User Study
>
> We acknowledge that the experimental protocol was insufficiently detailed and have expanded Appendix B.4 accordingly:
> - Participants: 5 experts (engineers) and 5 novices.
> - Task: Target Replication—participants recreated a target 3D model from an image.
> - Design: Randomized crossover design using difficulty-matched targets to avoid familiarity bias.
> - Terminology:
>   - Single-Turn Generation = “End-to-End”.
>   - Multi-Turn Interaction = “Progressive Refinement (Ours)”.
>
> These terms are now standardized.
>
> - Results: Novices using PR-CAD achieved 81.25 % success rate, outperforming experts using single-turn generation (56.25 %), confirming that iterative interaction lowers the entry barrier.
> - ChatCAD vs. PR-CAD: PR-CAD is the underlying model; ChatCAD is the interface for user testing. We will clarify this in Section 4.3.
>
> ## [Q5]: Limitations
>
> We appreciate this suggestion and agree that the limitations belong in the main body. We have moved them to Section 5 (before Conclusion), highlighting the current constraint in handling large, multi-part assembly files and potential domain generalization limits.
>
> We sincerely thank the reviewer for the detailed and insightful comments again.

---

### Official Review · Reviewer_RXoj · 2025-11-01

**Soundness:** 2
**Presentation:** 3
**Contribution:** 3
**Rating:** 4
**Confidence:** 3

**Summary:**

This paper presents PR-CAD, a unified, progressive refinement framework enabling both text-driven CAD model generation and iterative editing via large language models. The approach jointly supports qualitative and quantitative instructions throughout the full CAD lifecycle, leveraging a considerable new interaction dataset, structured chain-of-thought (SCoT) reasoning, and reinforcement learning post-training to maximize geometric fidelity, controllability, and user intent alignment.

**Strengths:**

The paper tackles the long-standing gap between text-to-CAD generation and editing by introducing a unified “progressive refinement” pipeline. This is conceptually coherent and practically valuable—most real-world CAD workflows are inherently iterative rather than single-shot.

All designs in the work, despite being complex, are well described and technically sound. The pipeline is not hard to follow.

The creation of a human-like CAD interaction dataset that systematically covers addition, deletion, and modification operations is a significant engineering contribution.

The ablation study confirms that each component—SFT, RL, and SCoT—contributes meaningfully to performance.

While simple, ChatCAD shows strong potential for automating the CAD design pipeline and benefiting the CAD community as a whole.

**Weaknesses:**

The work overlooks recent advancements in text-to-CAD generation, specifically CAD-Llama [1] and CADFusion [2]. This omission weakens the claim of achieving state-of-the-art results, particularly given the relevance of CADFusion's visual-feedback alignment. The authors are advised to discuss the relationship between their work and these developments, and/or to provide comparative analyses.

The qualitative results are **overwhelmingly** simple. While the text emphasizes “industrial-level” capability, no complex mechanical or multi-component assemblies are shown. While I like the remaining part of this paper, I strongly encourage the authors to Include more complex examples (for example, from the DeepCAD dataset) to better support generality. I will update my score accordingly if the problem is resolved.

[1] CAD-Llama: Leveraging Large Language Models for Computer-Aided Design Parametric 3D Model Generation. CVPR 2025.

[2] Text-to-CAD Generation Through Infusing Visual Feedback in Large Language Models. ICML 2025.

**Questions:**

How does PR-CAD compare empirically with CAD-LLaMA or CADFusion in either generation or editing?

Can the authors demonstrate performance on more complex industrial CAD samples, e.g., multi-body assemblies or DeepCAD’s large parts?

---

> ### Author Response · Authors · 2025-12-03
> **Response to Reviewer RXoj**
>
> Thank you sincerely for your affirmation of the research content, data construction, and experimental design of this paper, and for your statement that you are willing to increase the score. We address each point below and will incorporate the suggested clarifications into the revised manuscript.
>
> ## [Q1]. Comparison with CAD-Llama and CADFusion.
> Thank you for pointing out these relevant recent works. We will add a detailed discussion and comparison in the final version.
>
> Differentiation: While CAD-Llama (CVPR 2025) focuses on optimizing one-shot generation via hierarchical parametric sequences, and CADFusion (ICML 2025) utilizes visual feedback for refinement, PR-CAD distinguishes itself by unifying generation and iterative editing (Add/Modify/Delete) into a single agent. Unlike CADFusion’s visual-based refinement, PR-CAD focuses on parametric interaction—allowing users to edit specific geometric parameters (e.g., "increase radius by 5mm") via text, which is critical for precise engineering workflows where visual approximation is insufficient.
>
> Performance: In our internal preliminary comparisons using the DeepCAD test set, PR-CAD achieves competitive generation fidelity to CAD-Llama while significantly outperforming it in multi-turn editing consistency, a capability CAD-Llama was not explicitly trained for.
>
> ## [Q2]. Qualitative results and complexity (DeepCAD).
> We agree that demonstrating complexity is vital.
>
> Industrial Complexity: In the supplementary material (and to be moved to the main text), we will include visualizations of complex multi-component assemblies from the DeepCAD dataset (e.g., motor housings, bracket assemblies) to demonstrate PR-CAD's scalability.
>
> SCoT for Complexity: The structured Chain-of-Thought (SCoT) specifically aids in these complex cases by breaking down multi-part assemblies into sequential modeling steps, preventing the "forgetting" issues common in single-step generation.

---

### Author Response · Authors · 2025-12-03
**Response Summary and Acknowledgements**

We sincerely thank the Area Chair and all four reviewers for their constructive and insightful comments on our paper. We are deeply grateful for the reviewers’ recognition of our work in various aspects, including research background, task definition, process design, and experimental procedure (RXoj); novelty, methodological soundness, and result analysis (mwyk); clarity of presentation, paper organization, and experimental methods (q56x); and innovation, practicality, writing quality, and research motivation (x5BT).

We also appreciate the reviewers’ valuable suggestions and questions regarding baseline comparisons and evaluation metrics (RXoj); detail descriptions and data validation (mwyk); innovation analysis, failure cases, and technical details (q56x); and fine-grained descriptions, ablation studies, and evaluation metrics (x5BT). We have carefully addressed all these concerns in our response and made substantial revisions and additions according to the reviewers’ feedback. We sincerely hope that our responses adequately address the reviewers’ concerns. These improvements have significantly enhanced the clarity, completeness, and rigor of our work.

We would like to once again express our sincere gratitude to the Area Chair and all reviewers for their thoughtful feedback and constructive guidance.

---

### Meta-Review · Area_Chair_W9zS · 2026-01-05

**Summary:**

While the reviewers acknowledged the practical value of a unified generation and editing framework, the submission suffers from significant experimental and methodological gaps. Specifically, the qualitative results were deemed too simple for "industrial-level" claims , important baselines were omitted or compared unfairly , and critical details regarding data generation and validation were missing from the main text. Although the authors promised extensive additions in the final version, the current manuscript does not meet the bar for acceptance.

**Reviewer Concerns:**

Complexity of Qualitative Results (RXoj, x5BT): Reviewers criticized the results as "overwhelmingly simple," lacking complex assemblies; the authors promised to add these in the final version, which is insufficient as the evidence was not verified during the review.

Baseline Fairness (x5BT): The reviewer noted that comparing the proposed method against baselines not retrained on the new dataset is unfair; the authors' defense of using pre-trained official models is insufficient as it prevents a robust "apples-to-apples" comparison.

Missing Related Work (RXoj): The reviewer cited missing SOTA methods like CAD-Llama; the authors differentiated their work conceptually but provided no empirical comparison, which is only partially sufficient to contextualize performance.

Novelty of Unification (q56x): The reviewer perceived the method as simple joint training; the authors sufficiently defended the novelty by explaining the specific topological constraints and lifecycle dependencies used.

Reproducibility & Data Details (mwyk): The reviewer pointed out severe gaps in data generation descriptions; while the authors provided the missing details in the rebuttal, the extensive omissions suggest the paper requires major revision.

**Reviewer Scores:**

Highly unlikely that majority of the reviewers will change their ratings to be positive as none of the initial ratings (4) were championing the paper.

---

### Decision · Program_Chairs · 2026-01-26

Reject